# Smart scanning for low-illumination and fast RESOLFT nanoscopy in vivo

Jes Dreier [1], Marco Castello[2], Giovanna Coceano[1], Rodrigo Cáceres[3,4,5], Julie Plastino[3,4], Giuseppe Vicidomini [2] & Ilaria Testa[1]

RESOLFT fluorescence nanoscopy can nowadays image details far beyond the diffraction limit. However, signal to noise ratio (SNR) and temporal resolution are still a concern, especially deep inside living cells and organisms. In this work, we developed a non-deterministic scanning approach based on a real-time feedback system which speeds up the acquisition up to 6-fold and decreases the light dose by 70–90% for in vivo imaging. Also, we extended the information content of the images by acquiring the complete temporal evolution of the fluorescence generated by reversible switchable fluorescent proteins. This generates a series of images with different spatial resolution and SNR, from conventional to RESOLFT images, which combined through a multi-image deconvolution algorithm further enhances the effective resolution. We reported nanoscale imaging of organelles up to 35 Hz and actin dynamics during an invasion process at a depth of 20–30 μm inside a living *Caenorhabditis elegans* worm.

[1] Department of Applied Physics and Science for Life Laboratory, KTH Royal Institute of Technology, 100 44 Stockholm, Sweden. [2] Molecular Microscopy and Spectroscopy, Istituto Italiano di Tecnologia, Via Morego 30, 16136 Genoa, Italy. [3] Laboratoire Physico-Chimie Curie, Institut Curie, PSL Research University, CNRS, 75005 Paris, France. [4] Sorbonne Université, 75005 Paris, France. [5] Université Paris Descartes, 75005 Paris, France. These authors contributed equally: Giuseppe Vicidomini, Ilaria Testa. Correspondence and requests for materials should be addressed to G.V. (email: giuseppe.vicidomin@iit.it) or to I.T. (email: ilaria.testa@scilifelab.se)

Fluorescence nanoscopy[1] has potentially everything in its favor to become a fundamental tool for understanding the bio-molecular complexes that regulate life, because it can provide molecular spatial resolution while preserving the important assets of light microscopy such as live-cell and volumetric imaging. The transformative ability to image far beyond the diffraction limit is obtained by switching fluorescent molecules between two distinguishable states (ON−OFF)[2].

Stimulated-emission depletion (STED) nanoscopy[3,4] was the first viable concept using the ON−OFF concept. In STED nanoscopy, excitation and stimulated emission are used to switch the fluorescent molecules between the ground- (OFF) and the excited- (ON) states. However, because of the relatively short lifetime of the excited-state (~ns), the illumination intensity requested to effectively switch OFF a fluorescent molecule is ~GW/cm$^2$. Even if these intensities—in the far-red/near-infrared—are proven to be compatible with live-cell imaging[5], a valuable alternative has been obtained in REversible Saturable/Switchable Optical Linear Fluorescence Transitions (RESOLFT)[6–8] nanoscopy by exploring the photoinduced cis–trans isomerization of reversibly switchable fluorescent proteins (rsFPs)[7,9–11] or organic fluorophores[12]. Since the OFF state can reach milliseconds lifetime the illumination intensity needed by RESOLFT nanoscopy reduces to ~W-kW/cm$^2$. In a typical point-scanning RESOLFT nanoscopy architecture[10,13], distinct modulated light patterns are sequentially used to (i) switch ON the rsFPs within a diffraction-limited sized volume; (ii) switch OFF most of rsFPs except those located in a small subdiffraction volume, and (iii) read the residual rsFPs in the ON state via fluorescence. This complete RESOLFT cycle is repeated for each pixel and when applied to current available rsFPs leads to prolonged pixel dwell time on the order of 0.4–10 ms[7,14]. This dramatically slows down the imaging process, affecting the frame-rate in time-series live-cell imaging, especially for a large field of view with many recorded pixels.

Here, we propose a specimen-adaptive recording approach, which synergistically combines a real-time feedback system, modulated illumination and time-resolved acquisition to push the spatio-temporal resolution in RESOLFT nanoscopy and to further minimize the illumination doses. The time-resolved photon registration, synchronized with the RESOLFT imaging scheme, is used as input for a real-time feedback system that is able to spot the presence of labeled structures inside the illuminated area of the sample and to proceed with a smart and not predetermined scanning mode. Real-time feedback systems have been previously reported in confocal microscopy[15], to reduce light dosage, photobleaching and potential photo damaging effects, but the spatial resolution was still diffraction limited and the scan speed was constant. More recently, STED nanoscopy took advantage of real-time feedback to reduce the dose of sample illumination[16,17], but none of these approaches is adaptive in time or take advantage of the spatial information encoded in the fluorescence temporal evolution. Indeed, an important aspect of our specimen-adaptive recording is the ability to register the time-evolution of the fluorescence signal during the entire ON−OFF state transitions of rsFPs. This fluorescence emission is generated by distinct spatially and temporally modulated light patterns across the entire RESOLFT imaging scheme and gives rise to images with an extended spatial information content. The combination of such information through an algorithm based on multi-image deconvolution[18,19] enhances the signal -to -noise ratio (SNR) of the RESOLFT images, and thus its effective resolution. Our new way to probe and register the signal emitted by reversibly switchable proteins in a sample-matching fashion speeds up the recording time and minimizes the light exposure in RESOLFT nanoscopy, and we have named it smart RESOLFT.

We demonstrate the performance of smart RESOLFT nanoscopy by recording dynamic structures such as peroxisomes at 2–5 Hz for more than 100 frames, and by observing the fission and fusion dynamics of mitochondria at 27–40 Hz. Additionally, we acquire images of synaptic proteins in hippocampal neurons up to six times faster than in previous implementations. Finally, we record, for the first time deep inside a living nematode, 40 volumes of an actin-rich protrusion in an invading cell with both 80% reduced light exposure and subdiffraction spatial resolution.

These new strengths make scanning-based RESOLFT nanoscopy one of the best choices for long-term, subdiffraction spatial resolution and depth imaging of living scattering biological systems.

## Results

**Concept of specimen-adaptive recording in smart RESOLFT nanoscopy.** Conventional scanning microscopy is commonly based on a spatial and temporal predetermined raster scanning of one or more beams of light across the sample (Fig. 1). In this way each point of the sample is equally illuminated, even the regions that do not contain structures, which typically constitute a large portion (60–90% in all examples of this work) of the imaged area. This useless excess of illumination not only increases the chance of photo-toxicity but also lowers the acquisition speed. This is especially true in fluorescence nanoscopy, where many pixels need to be collected to ensure optimal sampling of nanoscale structures and where the spatial resolution and image contrast are only limited by the amount of available ON−OFF switching cycles and molecular brightness.

We developed a new smart scanning concept (Fig. 1a), which adapts to the specific features of the imaged sample. The rapid matching is based on a pixel-by-pixel decision and action, which is enabled by a feedback system based on a real-time controller implemented on a field-programmable-gate-array (FPGA) data-acquisition card (Fig. 1a). Towards the implementation of what we call "smart-pixel", the fluorescent photons emitted upon one or multiple probe-illuminations are recorded in multiple temporal channels/windows (Fig. 1b). In practice, the digital outputs of the FPGA can rapidly trigger pulses of different durations, colors and intensity focal distributions, i.e. Gaussian or doughnut-shaped. The fluorescence photons generated during these multiple temporal steps can be collected and stored accordingly in multiple channels (Fig. 1b). This flexible illumination scheme integrated with a rapid, intra-pixel, fluorescence signal classification is particularly powerful when applied to RESOLFT imaging, which is often characterized by continuous fluorescence photon emission during the ON−OFF-switching cycle. For instance, most of the current rsFPs used in RESOLFT nanoscopy emit a large number of photons before switching into the OFF state, since the fluorescence excitation light is coupled to OFF-switching. However, to our best knowledge, all previously reported RESOLFT imaging schemes discard the photon emission during the switching steps.

In our approach we take great advantage of this emission firstly to adapt the scanning to the presence of labeled structure, which speeds up recording and minimizes illumination (Fig. 1c) and secondly to increase the information content during recording (Fig. 1d).

If the number of photons during probing is lower than a defined threshold $T$ (negative pixel), the region in the sample will be skipped; on the contrary, if the number of photons is equal or higher than $T$ (positive pixel), the full RESOLFT illumination scheme will be applied. The threshold value $T$, as well as the shape, the wavelength and the duration of the probe beam should be carefully chosen in order to avoid scanning artifacts and false-

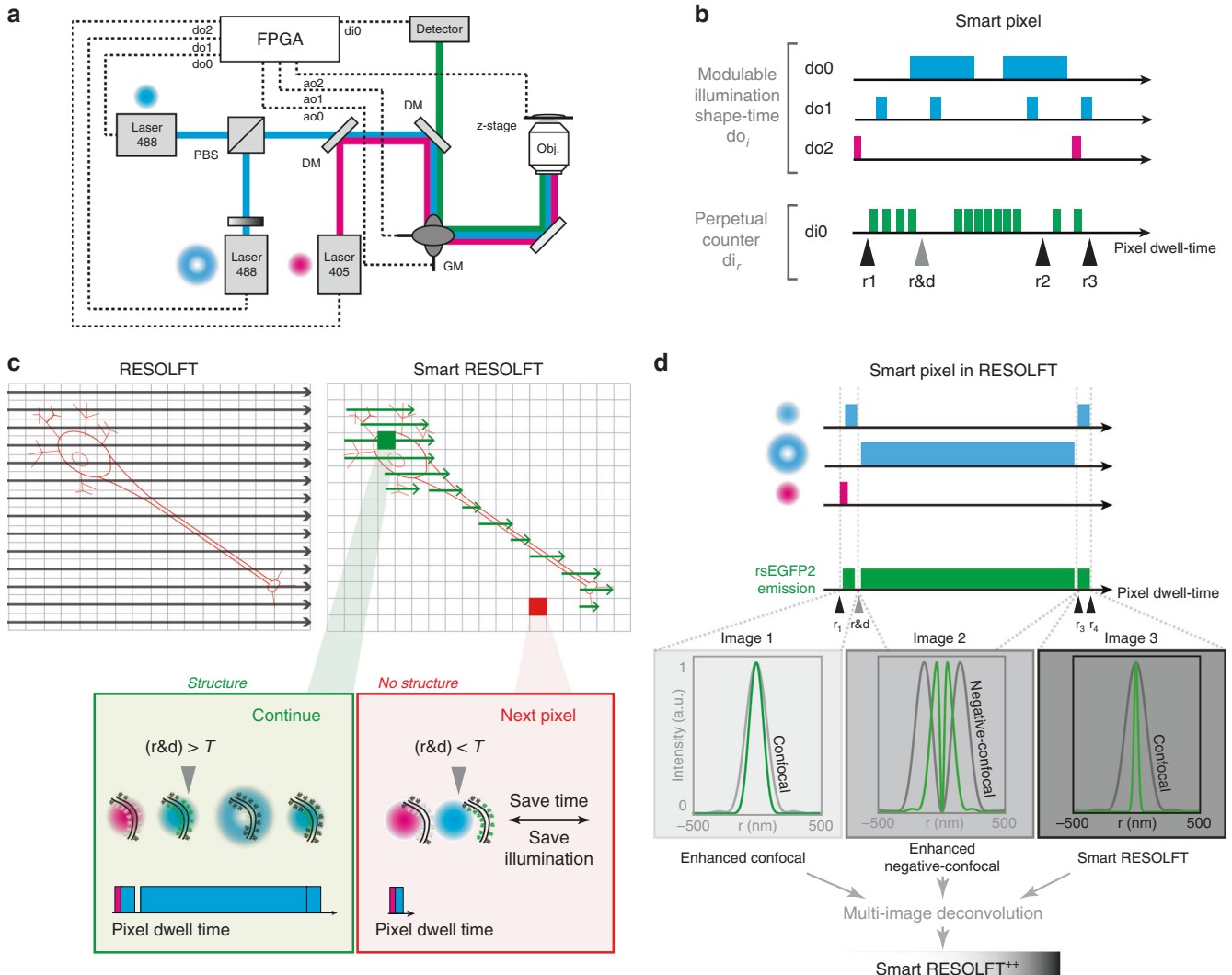

**Fig. 1** Smart pixel concept in RESOLFT imaging. **a** Hardware scheme of the smart RESOLFT sample-adaptive set-up. The field-programmable-gate-array card enables a rapid feedback loop between illumination, detection and scanning system. Multiple illumination lines, up to five digital outputs $do_i$ modulate up to five illuminations, which can change in shape and/or colors. The control software/hardware named Carma syncronizes the scanning architecture and the illuminations with three different analog output $ao_i$, and with the photon detection device $di_1$. **b** Schematic illustration of the flexible illumination and signal registration of Carma. The combination of real-time adaptive illumination and time binned detection is named smart pixel since it is applied on the very same sample region. The number and the duration of the illumination pulses and the width of signal classification windows can be flexibly adjusted. In particular, Carma implements a perpetual counter that is initialized at the start of the smart-pixel. Up to five different read-out points $r_i$ can be introduced within the deterministic pixel-dwell time. By calculating the difference between two consecutive read-out windows, it is possible to classify the photons collected under a specific illumination. The read-out points coincide with the activation/deactivation of the laser beams. The read-out window named r&d is introduced to implement the threshold-based detection for the real-time feedback system. **c** Raster (left) compared to smart (right) scanning, which is specimen-adaptive. The feedback loop and smart pixel technologies enable to perform RESOLFT imaging only in regions of the specimen containing labeled structures. The decision in smart RESOLFT (red and green boxes) is performed intra-pixel and it is threshold based. **d** Smart pixel in RESOLFT. The fluorescence photons are classified in three separated detection windows corresponding to the probing, doughnut (OFF-switching) and read-out illumination. In conventional RESOLFT only the windows corresponding to the read-out illumination ($r_4-r_3$) is recorded. Here, we combine the signal generated by the three windows through a multi-image deconvolution to generate an image of higher quality. RESOLFT REversible Saturable/Switchable Optical Linear Fluorescence Transitions

positive or negative pixels. Additionally, the smart scanning method registers the time evolution of the fluorescence signal in multiple channels in synchronization with the different illuminations. The multiple channels are successively used to reconstruct a high-resolution RESOLFT image via multi-image deconvolution.

**Probes and decision making.** The "smart pixel" concept allows to implement different probing schemes, which differ in complexity and robustness. Initially, we implemented probing

schemes that do not change the conventional sequence of RESOLFT; later additional pulses were added to improve the robustness of the method (Fig. 2).

In conventional RESOLFT imaging, the first illumination that probe the sample is the ON-switching light. However, the fluorescence signal emitted upon 405 nm illumination by green-emitting rsFPs such as DronpaM159T and rsEGFP2 is too weak to reliably probe the presence of structures. Alternatively, the feedback system can be based on the fluorescence induced by the

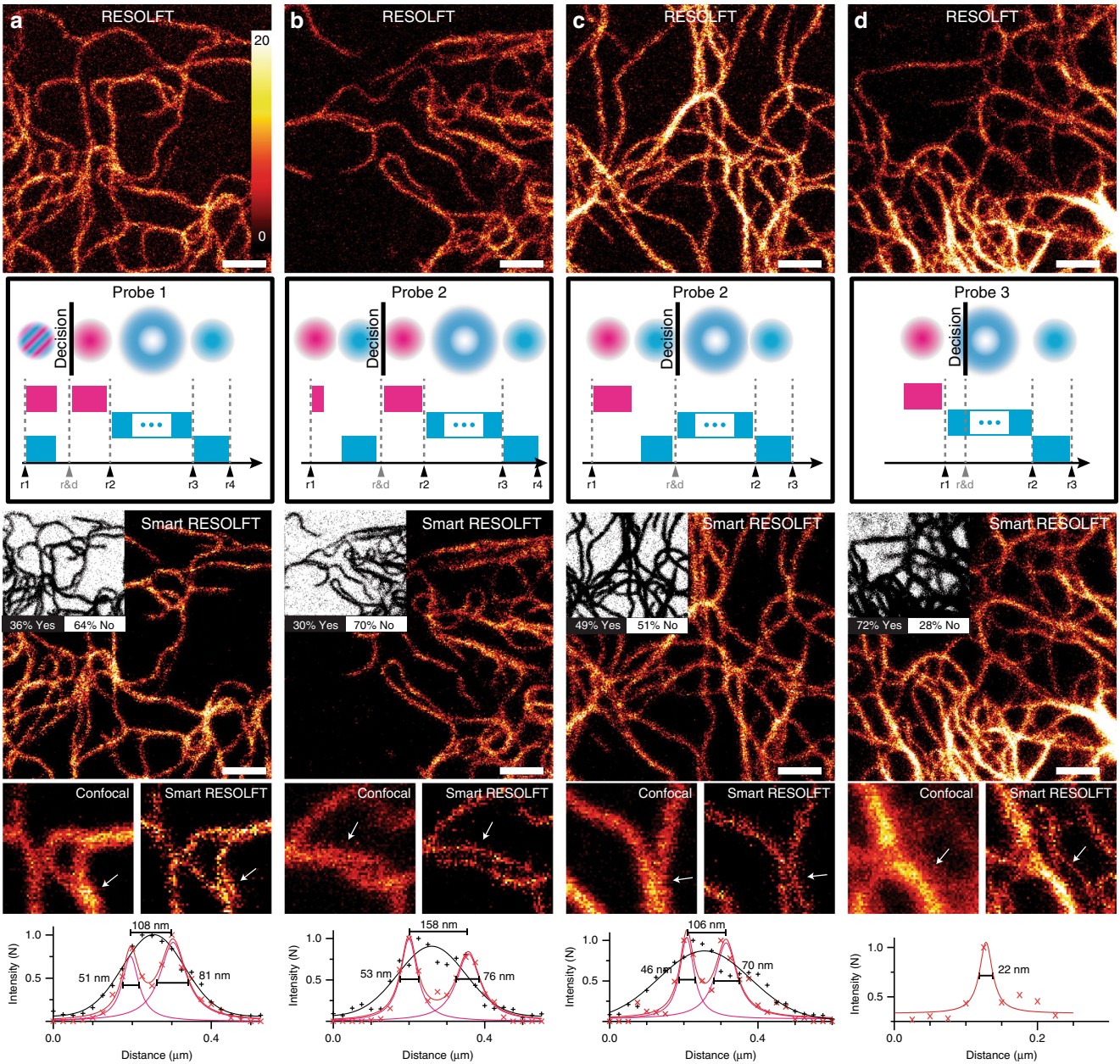

**Fig. 2** Probe types and decision-making in smart RESOLFT imaging. **a−d** In the top row conventional RESOLFT images of vimentin-rsEGFP2 are shown. By applying the smart RESOLFT illumination schemes illustrated below each image, we acquired a second image in the smart RESOLFT mode. The insets in the smart RESOLFT images represent the decision map containing the skipped pixels (white, NO), and the ones that undergo the full pulsing scheme (black, YES). For each smart RESOLFT image we provide a zoom of a small region, the related confocal image and a line profile (white arrow) for comparison. Each line profile is averaged across three pixels, and the full-width at half-maximum and peak-to-peak distance of Lorentzian fits is shown in the line profile. Scale bars are 1 μm, the zoom region is 1.5 by 1.5 μm$^2$. RESOLFT REversible Saturable/Switchable Optical Linear Fluorescence Transitions

OFF-switching doughnut-shaped beam at 488 nm (probe type 3, Fig. 2d). Here, the fluorescent signal is stronger; thereby, a short temporal window can be used to take the decisions. On the other hand, the probed region in the specimen is larger due to the intensity distribution of the doughnut and does not include the doughnut "zero" intensity point, which can lead to false positives (Supplementary Note 2 and Supplementary Figure 1).

The possibility to temporally modulate all beams of a conventional RESOLFT nanoscope allows for the implementation of more robust decision schemes without any modification on the optical system. For example, the probe type can include the 405 and 488 nm Gaussian beams applied simultaneously (probe type 1, Fig. 2a) or in temporal succession (probe type 2, Fig. 2b, c). In

this case the probed area is reduced due to the double confinements of the ON-switching and excitation. Since the goal of 488 nm Gaussian probe beam is to detect the presence of labeled structures in a certain sample region, its duration should be short enough to minimize light exposure but also able to generate a fluorescence signal, distinct from the background, to allow for robust decisions. A subsequent 405 nm ON-switching Gaussian beam step for the positive pixel can be used to again switch ON the rsFPs that were switched OFF during the probe step (probe type 2 with re ON-switching, Fig. 2b).

We compared the performance of different probing types by imaging vimentin structures in knock-in cell lines[20] expressing an rsEGFP2 fusion protein at endogenous levels. Here, the minimal

fluorescence variability from cell-to-cell guaranteed a reliable comparison for both spatial resolution and contrast between images acquired in different modalities. We recorded, in the same region of the sample, a series of eight images in the following modes: confocal (without OFF-switching), conventional (without decision), smart RESOLFT with probe type 1–3 then again conventional and confocal for comparison. We repeated the experiments by changing the order of the probe types to exclude the influence of photo-bleaching. Three independent samples were imaged. The imaging parameters and the results of the comparison are shown in Supplementary Tables 1 and 2.

Four representative experiments are shown in Fig. 2. The images acquired with the conventional scanning and the smart RESOLFT modes show comparable contrast, spatial resolution and content without any detectable loss of information.

This indicates that smart RESOLFT skips useless pixels without affecting the quality of the super-resolution images. A statistical measurement of the filaments profile is shown in Supplementary Figure 1. The four investigated probe types led to images with comparable spatial resolution (Supplementary Note 1). The image contrast is slightly worst for probe type 2 because it did not include an additional ON-switching step of the rsFPs.

As expected when using probe type 3 we illuminated more false-positive pixels (Supplementary Note 2 and Supplementary Figure 1), because the donut-based probe type measures a region that is wider and therefore less precise than the Gaussian-based. On the other hand, probe type 3 has the advantage of the simplest implementation and the gentler decision-making since it does not require additional beams.

Probe types 1 and 2 with re-switching led to images with the most precise decision map, the fastest recording time and the best combination of image contrast and spatial resolution.

**Time-resolved acquisition and image restoration**. The fluorescent photons registered during the smart RESOLFT cycle and recorded simultaneously encode different spatial information (Supplementary Note 3 and Fig. S2). The first image acquired contains the photons emitted during the Gaussian 488 nm probe (Fig. 3), which shows enhanced spatial resolution with respect to the confocal counterpart due to the sequential double modulation of the ON-switching and fluorescence excitation[8,21]. It is important to observe that the resolution enhancement is achieved when all the proteins are populating the OFF state when starting the RESOLFT pixel cycle, which is practically true in all our experiment schemes. The second image is named enhanced negative image and it is generated by the photon emitted under the illumination of the donut-shaped 488 nm OFF-switching. The fluorescence signal in this image is also the result of the double modulation of the ON-switching and the fluorescence excitation (Supplementary Figure 2). The final image is then composed by the photons acquired during the illumination of a 488 nm Gaussian-shaped beam, which is applied after OFF-switching. Thus, the photons are generated by a subdiffraction region in the sample and directly lead to a raw RESOLFT image. The difference in spatial resolution and contrast of the three images can be observed and compared to a true confocal image. The enhanced confocal showed a clear increase of spatial resolution, which has been carefully investigated (Supplementary Figure 3). Resolution further improves in the raw RESOLFT image, thanks to the confinement of the fluorescence emission, but it is accompanied by a reduction of SNR which may deteriorate the effective resolution. We solved this problem by merging the spatial information contained in sequentially recorded images (enhanced confocal, enhanced negative image and RESOLFT raw) with a multi-image deconvolution algorithm. This way, we re-assign all the photons to the most-likely original emission location with high fidelity, which generate a single high-resolution RESOLFT[++] image. The effective resolution enhancement is also confirmed by a Fourier ring correlation analysis[22] (Supplementary Note 4-6 and Supplementary Figure 4).

A common concern in STED and RESOLFT imaging of dim samples is to miss structures because of photo-bleaching during imaging. The signal classification that comes with our smart RESOLFT concept allows to diagnose this problem by providing, in a single measurement, a direct comparison between the enhanced confocal and RESOLFT approach and to further minimize its occurrence by actively reducing photo-bleaching.

**Live cell imaging**. Neuronal cells cover very large areas due to their highly polarized morphology, but only occupy a fairly small amount of this area with their processes. Therefore, when imaging neurons typically a large portion of the image is composed by dark pixels and does not contain any valuable information. We undertook a structural investigation of the neuronal protein Synapsin tagged with DronpaM159T, which is associated with the clustering of synaptic vesicles in the synaptic bouton[23]. The labeled structures typically occupied less than 6% of the total amount of pixels, thus greatly benefiting from a nondeterministic scanning approach, which led to the usage of 90% less light and six times faster recording time compared to conventional RESOLFT imaging (Fig. 4).

Each image showed clusters as well as small individual synaptic vesicles with diameters of 50–70 nm, which were not resolvable in the related conventional images. Furthermore, we used the algorithm above to enhance the images resulting in the smart RESOLFT[++], which further highlights the individual vesicles and helps to better distinguish the clusters by increasing the SNR.

The power of smart RESOLFT to decrease the acquisition time and lower the photo dosage can also be used to improve the capabilities of RESOLFT to capture dynamic processes of organelles (Fig. 5). Peroxisomes are highly mobile organelles, which serve a critical role in cells, and are involved in processes such as oxidation of fatty acids, cholesterol and bile acids among others. Peroxisomes defects lead to many different diseases[24]. Despite the importance of peroxisomes, their function and mechanism are not fully understood, and their biogenesis in mammalian cells remains largely unknown[25]. We investigated PEX16, a peroxisomal membrane protein required for peroxisome membrane biogenesis. Due to the mobile nature of peroxisomes, a lot of previous studies have either been performed on fixed cells with super-resolution nanoscopy, where individual peroxisomes could be investigated[26], or have been imaging the bulk of the cell, using conventional microscopy where individual peroxisomes cannot be distinguished. The minimal light exposure and the faster frame-rate enabled by our new smart RESOLFT imaging mode allowed us to follow the continuous interplay of single organelles for over 100 frames at 2–5 Hz (Fig. 5a, b). The spatial resolution and the speed of the smart RESOLFT enables us to capture the individual peroxisomes and track them from frame to frame. The recording speed at 2–5 Hz minimized motion artifacts, which would typically be encountered when imaging dynamic and freely moving vesicles using a scanning system with long line to line intervals (Supplementary Figure 5 and Supplementary Note 7). Furthermore, the increased spatial resolution enabled us to distinguish and separate individual peroxisomes, which allowed us to study the behavior and fate of individual peroxisomes. For instance, we distinguished a fusion event of two peroxisomes moving close to each other (supplementary Fig S5). In 100 frames, we recorded 24 trajectories (Fig. 5b) consisting of 41 individual points, in which we observed

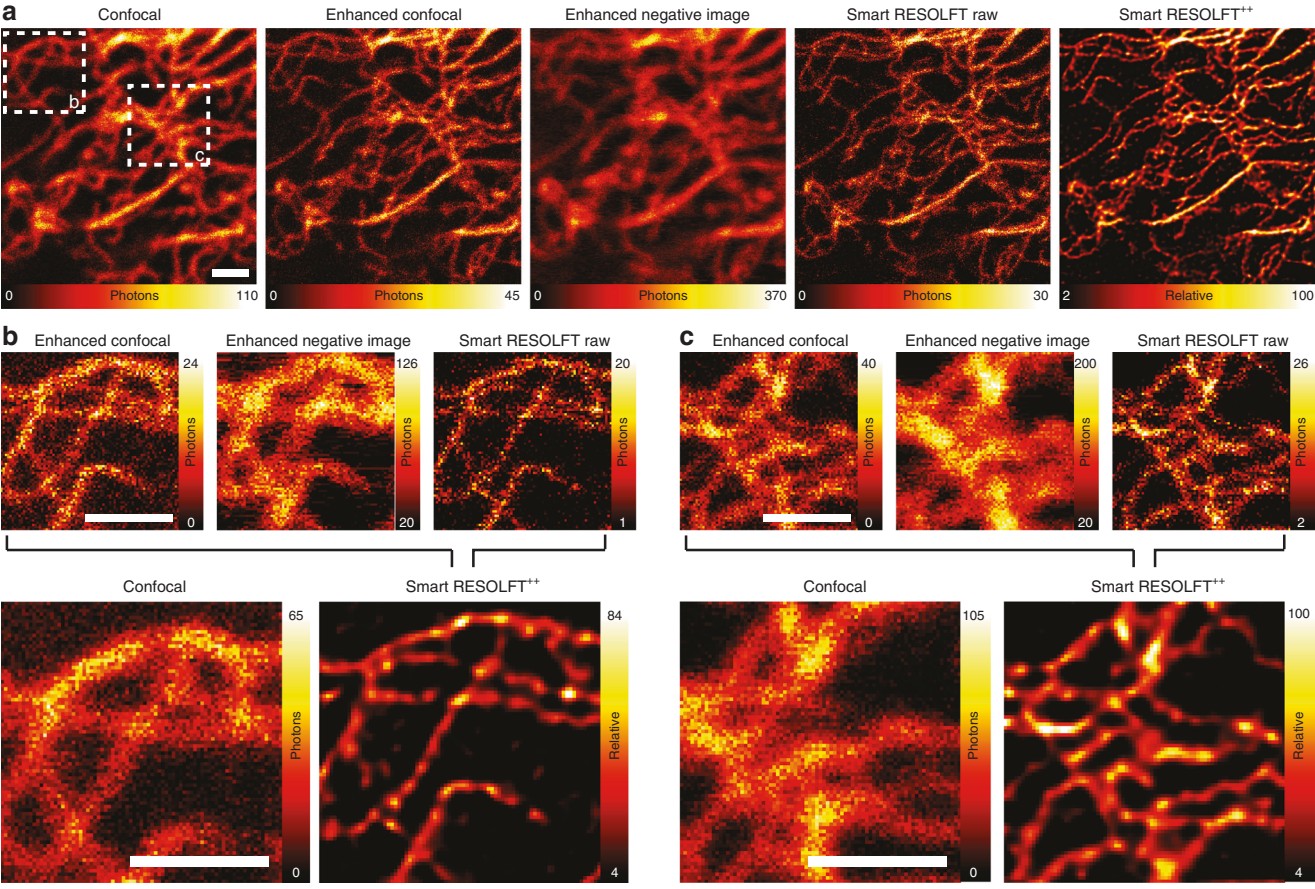

**Fig. 3** Extended information content in smart RESOLFT single recording. **a** The enhanced confocal, the enhanced negative image and the smart RESOLFT are all recorded in a smart pixel by a single scan of the sample. Each structure of the sample is illuminated according to probe type 2. The images are then combined to form the smart RESOLFT++ image through a multi-image deconvolution (five iterations). Included for comparison is also the confocal image. **b**, **c** Zoom of two regions of interest, which highlight the increased spatial resolution of the smart RESOLFT and smart RESOLFT++ images in comparison to the negative confocal, confocal and enhanced confocal images. Scale bars are 1 μm. RESOLFT REversible Saturable/Switchable Optical Linear Fluorescence Transitions

several fission and fusion events (supplementary Fig S5). Many of the trajectories were short with only 3–10 frames before the peroxisomes moved out of the frame. The regions in the image where peroxisomes clustered gave rise to longer trajectories with typically a slower speed. Using the trajectories, we measured a mean speed of about 0.44 ± 0.33 μm/s for $N = 433$ steps (Supplementary Figure 6, Supplementary Note 8). The ability to resolve and follow individual peroxisomes as in smart RESOLFT could be valuable to further investigate peroxisome biogenesis.

In another proof of concept, we investigated the dynamics of mitochondria by imaging OMP25-rsEGFP2-labeled membranes. We followed the overall dynamics of the mitochondria and their contact points with smart RESOLFT in a human cell line (Fig. 5c, d). The outer membrane of distinct mitochondria was often observed to come into close proximity before the formation of contact points or even fusion (Fig. 5c, inset). Most of the mitochondria were very mobile (colors in Fig. 5d) despite a small localized fraction (white, Fig. 5d). To further investigate the mitochondrial membranes' dynamics in real time, we recorded a line scan across the mitochondria, where the two membranes are well visible and separated (Fig. 5e). The line scan was recorded in the same region at 27–40 Hz for about 9600 time points (Fig. 5e, and Supplementary Figure 6).

The range of speed is due to the fact that the smart RESOLFT recording time was specimen-adaptive, but overall remained 50% faster than conventional RESOLFT. We observed the mitochondria dividing and fusing back together again multiple times thanks to the high time and spatial resolution, but also to the SNR that allow thousands of recordings of the same region (Fig. 5e). The time interval from fission to fusion were 10.1 and 17.1 s respectively. During 312 s a total of 4 such processes were observed, and in all cases, fission was only transient and followed by a fusion process.

**In vivo imaging.** Among the benefits of point scanning RESOLFT nanoscopy is the possibility to observe structures deep inside tissues and organisms thanks to a confocalized scheme of lenses, the presence of a pinhole in the detection and the confined fluorescence emission of the rsFPs. The smart RESOLFT's temporal resolution and minimal photo-toxicity allowed in vivo imaging of the actin dynamics of a single cell, the anchor cell, in the developing vulva of the *Caenorhabditis elegans* worm. The anchor cell is specialized in breaching an extracellular matrix barrier known as basement membrane, in a process called invasion, which is important in organ development and in cancer cell metastasis[27]. Anchor cell invasion takes place deep inside the worm (20–30 μm) over about 1 h and involves fast shape changes of the actin structures that push out the front of the anchor cell[28].

Using Lifeact-DronpaM159T, expressed only in the anchor cell, we imaged the structure and dynamics of filamentous actin

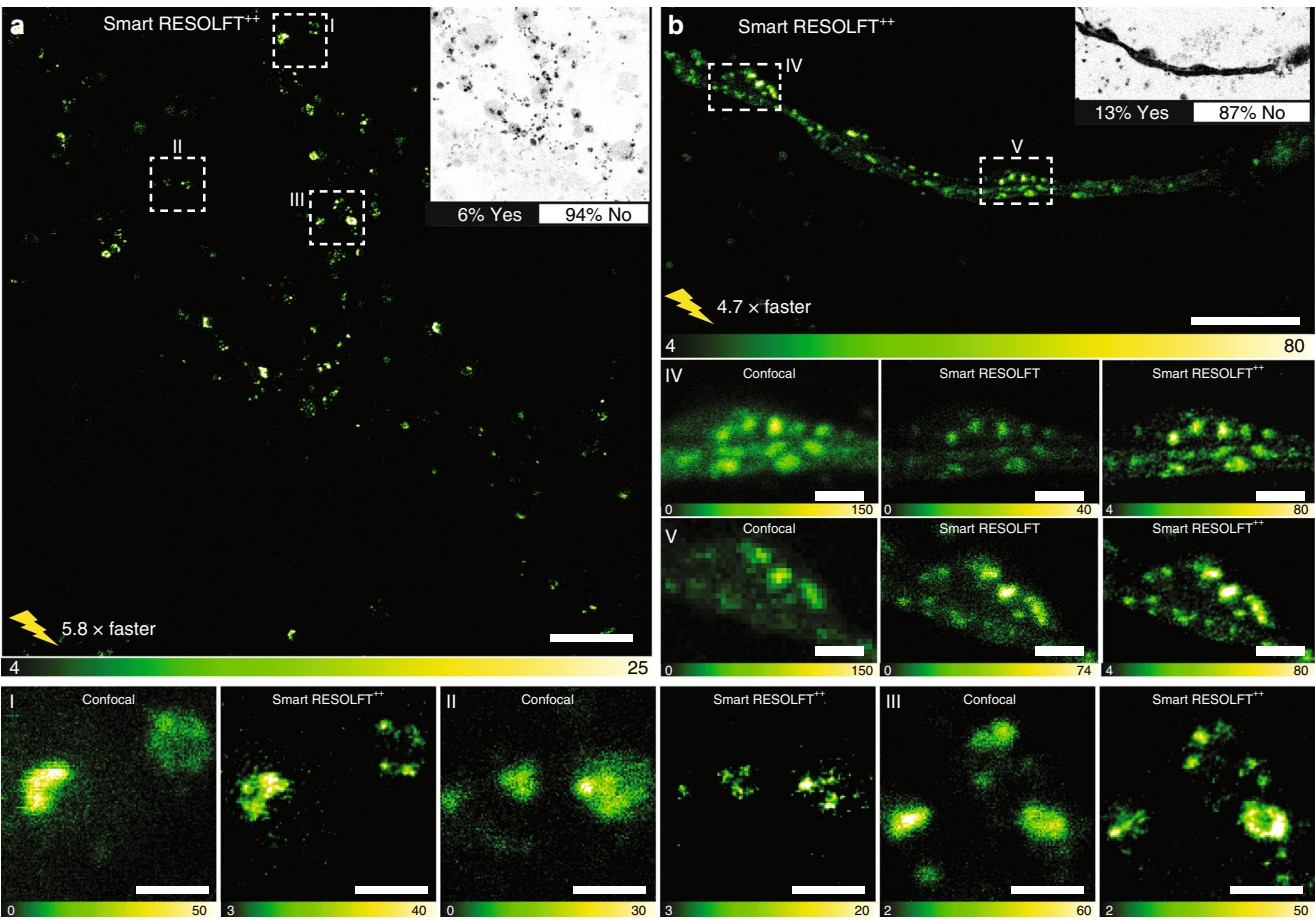

**Fig. 4** Smart RESOLFT imaging of hippocampal neurons. **a**, **b** Smart RESOLFT images of Synapsin1A-DronpaM159T, using probe type 2. The images were recorded 5.8 and 4.7 times faster than in conventional RESOLFT imaging with a reduced light dosage of 90% and 83% respectively. The insets show the decision maps with the percentage of skipped pixels (white, NO) and the fully illuminated pixels (black, YES). FOV of 40 by 40 µm² (**a**), and 30 by 15 µm² (**b**). I−III and V−IV region of interest (ROI) of images (**a**) and (**b**) respectively, showing single synaptic vesicles and small clusters. Scale bars are 10 µm (**a**), 5 µm (**b**), and 1 µm in ROI I−V. RESOLFT REversible Saturable/Switchable Optical Linear Fluorescence Transitions

in the large anchor cell protrusion that forms during basement membrane breaching and then retracts once a large basement membrane hole has formed (Fig. 6). To minimize miss-matching in the refractive index when imaging deep inside the worm, we used a glycerol objective lens[13]. We imaged a volume of 256 µm³ to observe the entire anchor cell in about 60–80 s, which is less than a third of regular RESOLFT recording. Only illuminating the portion of the anchor cell containing labeled structures allowed us to record 40 volumes in about 45 min. The adaptive illumination also decreases the light dose by 72% compared to conventional RESOLFT, which allowed to minimize the light hitting the worm. To depict the actin protrusions formed in the three-dimensional structure of the anchor cell, we show each volume from time-points 1 to 9 and also time-point 25 as maximum image projection with a color code corresponding to the relative depth (Fig. 6b). In volume 1, the actin protrusions were large and dynamic, exhibiting lateral spike structures that appeared and disappeared over the course of the first nine volumes. The cell underwent a remarkable reduction in dynamics and size from the starting point of the measurement to volume 25 and this became even more pronounced when observing volume 40 (Fig. 6c) where the protrusions were almost completely retracted. This was quantified (Fig. 6d) and the number of new spikes appearing was observed to decrease over the first 20 time-points, while the lifetime of existing spikes increased. In comparison to conventional RESOLFT, the smart RESOLFT approach allowed the recording of 26 more volumes with a much lower light dosage for a fixed recording time of 45 min (Fig. 6e).

## Discussion

So far, the possibility to obtain subdiffraction imaging with minimal sample perturbation and in vivo is the most important strength of RESOLFT nanoscopy (Supplementary Note 10). Here, we enhanced RESOLFT imaging by introducing a new scanning and signal classification concept, the so called "smart pixel", which led to up to 90% reduction in light doses together with time-lapse imaging at higher frame-rate compared to previous RESOLFT nanoscopy implementations. Additionally, the smart use of detected photons only accessible with our novel time-resolved acquisition led to images with an extended information content and higher SNR.

We demonstrated the robustness of our approach by imaging vimentin filaments and synaptic vesicles with a spatial resolution of 40–60 nm (full-width at half-maximum of Gaussian fitted line intensity profile). Rapidly moving organelles such as peroxisomes or mitochondria can be tracked inside the cells at 4–30 Hz for hundreds or even thousands of time points. The smart RESOLFT advances enabled for the first time in *C. elegans* in vivo super-resolution volumetric imaging. We tracked the actin dynamics of a single invading cell inside a living organism over 40 volumes (40

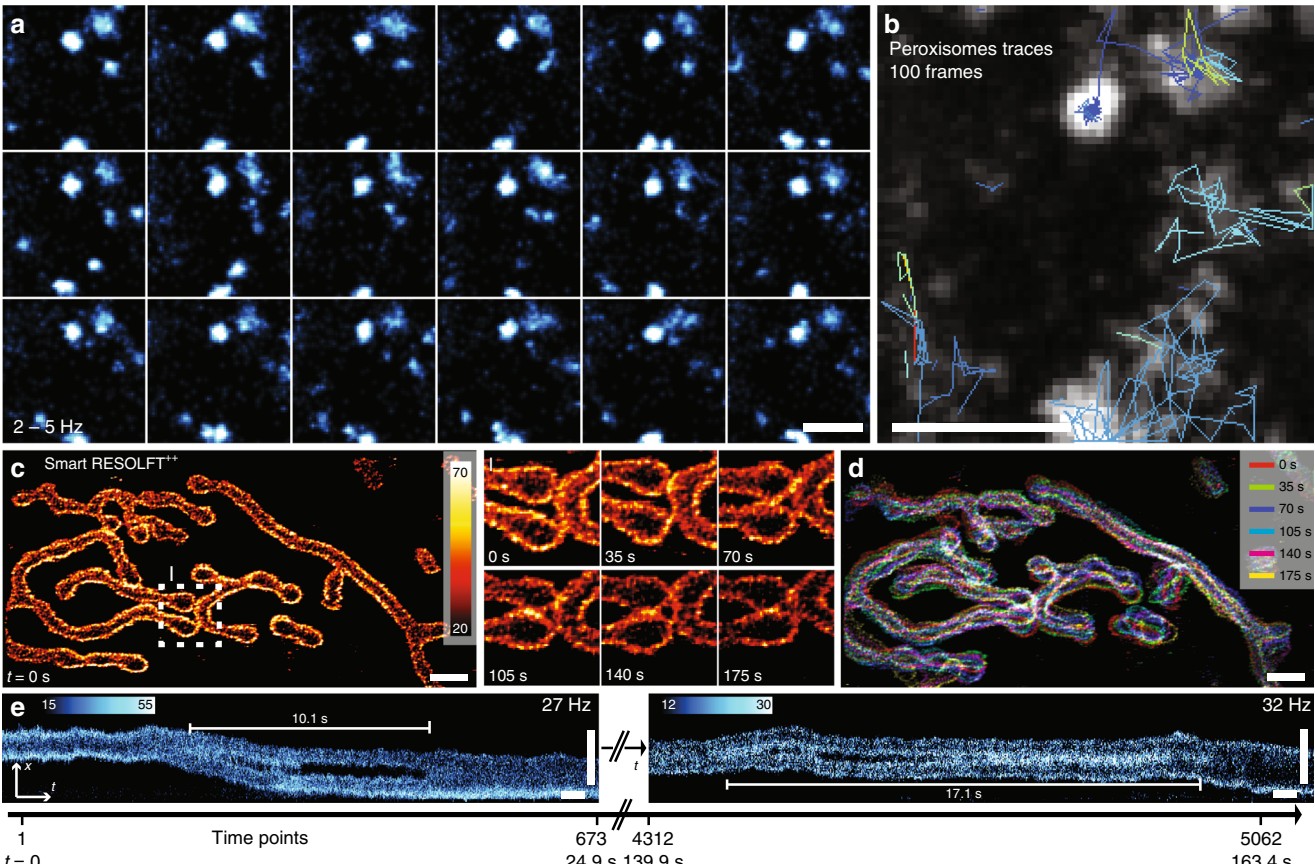

**Fig. 5** Rapid time-lapse imaging of organelles with smart RESOLFT. **a**, **b** We imaged the continuous interplay of peroxisomes over 100 time points recorded at 2–5 Hz. Pex16-rsEGFP2 was expressed in U2OS cells. The decision was made with probe type 1. **a** The first 18 frames of the time lapse are shown. Each image has been smoothed with a 30 nm radius Gaussian filter. **b** Maximum projection of the entire 100 frames time-lapse. Few selected peroxisome trajectories are shown in different colors. **c**, **d** Mitochondria dynamics in U2OS cells. The mitochondrial outer membrane was labeled with rsEGFP2 with a targeting sequence of Omp25. **c** Time lapse recorded in the smart RESOLFT mode. The first frame is shown in the fire color bar. The related region of interest ROI I shows mitochondrial membranes fusion over time. **d** Maximum projection of the time-lapse mitochondria imaging. Each image is color coded and identify the fraction of mitochondria with minimal mobility (white) from the mobile one (colored) in the seconds time frames. **e** To detect real-time (30 Hz) dynamics of the individual mitochondria we performed a line recording xt-scan over thousands of time points. We observed sequential fission and fusion of the outer mitochondrial membrane with sub-80 nm spatial resolution. Scale bars are 1 μm (**a**−**d**), for (**e**) the vertical scale bar is 1 μm and the horizontal 1 s. RESOLFT REversible Saturable/Switchable Optical Linear Fluorescence Transitions

time-points, corresponding to about 45 min) with subdiffraction spatial resolution. Currently, fluorescence optical nanoscopy techniques based on point scanning can image at the highest penetration depth with unique potential for tissue and in vivo imaging. Here we pushed the temporal aspect of point scanning with the development of a new specimen-adaptive interface, which can be easily applied to new rsFPs and combined to functional intra-pixel lifetime imaging[14], since the doughnut elicited fluorescence emission is substantial. Also, other techniques such as STED nanoscopy could benefit from our rapid signal classification and multi-image deconvolution algorithm to enhance the image quality. In this context it is interesting to compare our smart RESOLFT with the DyMIN STED nanoscopy implementation[16]. Despite the fact that it is applied to different types of state transition, the DyMIN concept does not modify the frame-rate and does not enhance the fluorescence signal emitted during the decision illumination. However, the DyMIN strategy continuously probes the presence of structures in a spatially dynamic way, which helps to achieve a desired spatial resolution without using a constant maximum STED intensity. Similar strategies can be included also in our "smart-pixel" technology;

however, since in RESOLFT the intensities are already moderate, this feature does not provide a large benefit. Finally, the smart RESOLFT sample-matching dwell time can be generalized to any scanning techniques spanning from two-photon, confocal as well as STED to speed-up the recording time, especially in combination with faster scanning technology. Overall, our smart scanning concept expands the spectrum of applications of live-cell nanoscopy in physiologically relevant systems such as tissues, 3D matrix and multi-cellular organism towards innovative questions at the small organelles and macromolecular complexes level.

## Methods

**Imaging acquisition**. The images were acquired using a beam scanning RESOLFT setup. Three modulable CW laser (Cobolt, Sweden) were combined into a XY galvo scan setup (Cambridge Technology, USA), and further directed into a Leica stand with a ×100 oil NA 1.4 objective (or ×63 Glyc NA 1.3 for the *C. elegans* images). The emission photons where de-scanned and collected with a SPAD (MPD, Bolzano, Italy). The stage was custom-built and mounted directly onto the objective with a built-in z-piezo. The xy-galvo's, the z-piezo and the pulsing scheme of the lasers were controlled using an FPGA, and a custom designed software. A more detailed description and a schematic can be found in the supplementary documents (Supplementary Note 9 and Supplementary Figure 7). The image acquisition parameters are reported in Supplementary Table S1.

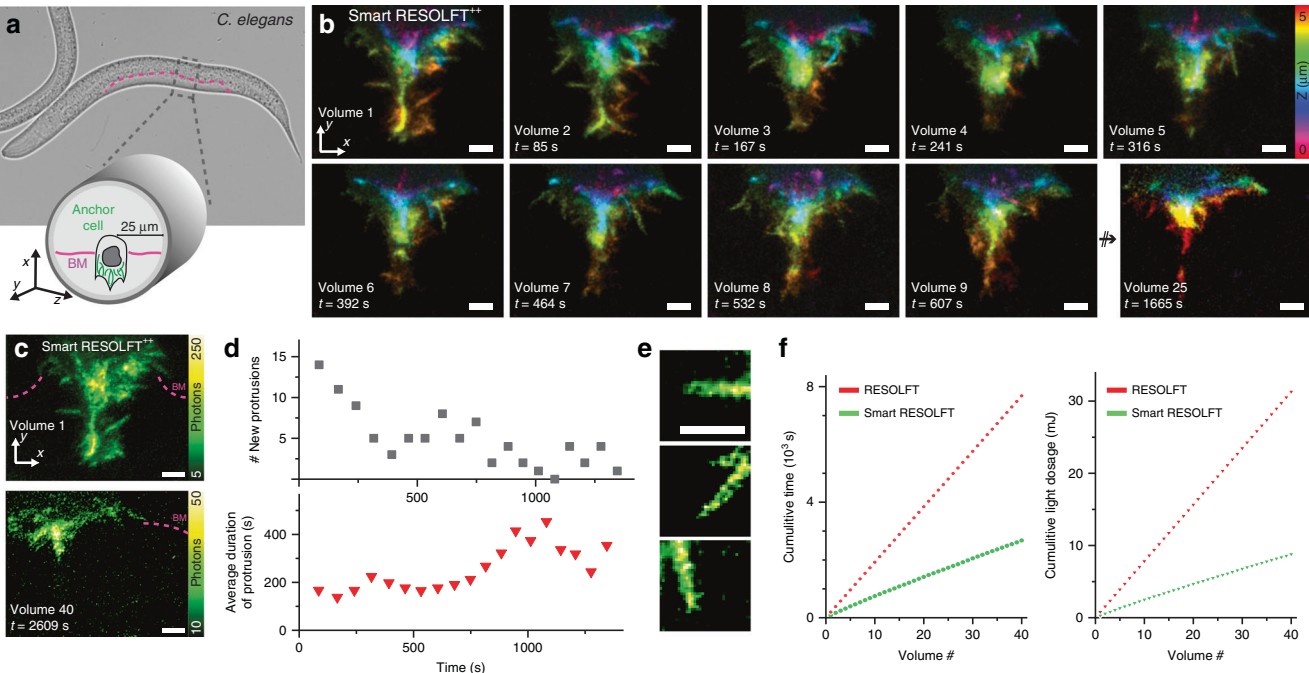

**Fig. 6** Smart RESOLFT imaging in vivo. **a** Conventional transmitted image of *C. elegans* worm before invasion, with a cartoon showing approximate anchor cell location and anchor cell shape during late invasion. The cell is expressing Lifeact-DronpaM159T which labels actin filaments (depicted in green) and the basement membrane that is being invaded is depicted in magenta. **b** We recorded 40 volumes of the anchor cell protrusion post-invasion when it has reached its maximum size and has started retracting. The recording was continuous over 40 min to follow the actin rearrangement post-invasion. Ten representative volumes are shown as color-coded maximum projection of the axial position. Each volume consists of ten smart RESOLFT images. The decision was made with probe type 2. **c** Maximum intensity projections of first and last (40th) recorded volumes to highlight the retraction stage of invasion. By the time of the last recorded volume, the anchor cell protrusion was mostly retracted as per usual for this stage of morphogenesis. **d** Plots displaying the number of actin spikes and their lifetimes during protrusion retraction post-invasion. At the beginning of the process, newly formed actin spikes are more numerous (top) but dynamic and short-lived (bottom), while at end of the acquisition when the protrusion is retracting, new spikes are rare, but existing spikes are more stable (longer lifetimes). **e** The images show three representative protrusions with fine actin bundles. They are examples of the protrusions detected and measured in the graphs. **f** The two graphs show the reduced exposure time (left) and light doses (right) of the smart RESOLFT (green) compared to conventional RESOLFT (red) recording. Scale bars are 1 μm

We implement the "smart-pixel" function into the data-acquisition/control/processing software Carma, which uses a data-acquisition (DAQ) card based on an FPGA. In particular, the microscope controller Carma (which include both the hardware and the software) is implemented on a commercial National Instruments NI PCIe-7852 card, equipped with a Virtex-5 FPGA chip. The card is used to synchronize the scanning system (galvanometer mirrors and z-piezo) with the laser beams modulation and the photons detection. Up to five digital synchronized output signals can be used to modulate the intensities of five different laser beams. Thanks to the implementation of a perpetual photon counter initialized at the beginning of the deterministic dwell time and different (up to five) read-out points (the counter is read), it is possible to register the number of photons collected within specific temporal windows synchronized with the intensity beams modulation. One read-out and decision point is introduced to implement the real-time feedback system. The photons collected at the decision point and a fixed threshold value allows to decide if the current pixel contains a structure or can be associated to the background. The read-out points allow to perform the signal classification and generate the images with different information content, such as the enhanced confocal, the negative RESOLFT and the smart RESOLFT images.

**Multi-image deconvolution.** A multi-image deconvolution algorithm is directly implemented on the Carma software to fuse the different images collected within the different time-windows into a single high SNR and high-resolution nanoscopy image. The multi-image deconvolution algorithm is an extension of the well-known Richardson–Lucy algorithm[29] routinely used in the context of fluorescence microscopy and super-resolution nanoscopy[18,19]. The algorithm is iterative and reads (Supplementary Note 5-6))

$$\mathbf{x}^{k+1} = \mathbf{x}^k \left( \sum_{l=1}^{L} w_l^{-1} \mathbf{h}_l \star \frac{\mathbf{y}_l}{\mathbf{h}_l * \mathbf{x}^k} \right),$$

where $\mathbf{y}_l$ and $\mathbf{h}_l$ are the different images and the associated PSFs, $\mathbf{x}^k$ the

reconstructed smart RESOLFT image at iteration $k$ and $w_l$ a weight factor able to consider the difference in brightness of the different images. The correlation operator $\star$ and the convolution operator $*$ are implemented through fast Fourier transforms.

**Tracking of the peroxisomes.** The tracking of the peroxisomes was carried out using the Fiji plugin TrackMate[30].

**Epithelial cell culture and rsFPs plasmids.** U2OS (ATCC® HTB-96™) cells were cultured in Dulbecco's modified Eagle's medium (DMEM) (Thermo Fisher Scientific, 41966029) supplemented with 10% (vol/vol) fetal bovine serum (Thermo Fisher Scientific, 10270106), 1% penicillin-streptomycin (Sigma Aldrich, P4333) and maintained at 37 °C and 5% CO₂ in a humidified incubator. For transfection, $2 \times 10^5$ cells per well were seeded on coverslips in a six-well plate. After 1 day cells were transfected using Lipofectamine LTX with PLUS reagent (Thermo Fisher Scientific, 15338100) according to the manufacturer's instructions. 24 to 36 h after transfection cells were washed in phosphate-buffered saline solution, placed with phenol-red free DMEM or Leibovitz's L-15 Medium (Thermo Fisher Scientific, 21083027) in a chamber and imaged at room temperature. The plasmid rsEGFP2-OMP25 was cloned as described in Masullo et al.[8]. The rsEGFP2_Pex16 was a kind gift from Dr. Stefan W. Hell and Dr. Stefan Jakobs (MPI-BCP Göttingen, Germany).

**Hippocampal neuron cultures and infection.** Primary hippocampal neuron cultures were prepared from embryonic day E18 Sprague–Dawley rat embryos. Hippocampi were dissected and mechanically dissociated in MEM (Minimum Essential Medium, Thermo Fisher Scientific, 11095080). $20 \times 10^3$ cells per well were seeded in 12-well plates on a poly-D-ornithine (Sigma Aldrich, P8638)-coated #1.5 18 mm (Marienfeld) glass coverslips and let them attach in MEM with 10% horse serum (Thermo Fisher Scientific, 26050088), 2 mM L-Glut (Thermo Fisher

Scientific, 25030–024) and 1 mM sodium pyruvate (Thermo Fisher Scientific, 11360–070). After 3 h the media was changed with Neurobasal (Thermo Fisher Scientific, 21103–049) supplemented with 2% B-27 (Thermo Fisher Scientific, 17504–044), 2 mM L-Glutamine and 1% penicillin-streptomycin. The experiments were performed on cultures starting from DIV14. 12–24 h before experiments, the cells were infected with a modified Semliki Forest Virus expressing either the actin binding protein Lifeact or the synapsin1A protein together with the photo-switchable protein DronpaM159T, by adding 1 μl of the virus to the culture medium. All experiments were performed in accordance with animal welfare guidelines set forth by Karolinska Institutet and were approved by Stockholm North Ethical Evaluation Board for Animal Research.

**C. elegans preparation and culture**. Worms were maintained and handled using standard techniques[31]. For time-lapse microscopy, worms were anesthetized in 0.02% levamisole in M9 solution and then transferred to 4.5% noble agar pads, sealed with VALAP, and imaged at 20 °C.

We designate linkage to a promoter with a greater than symbol (>) and use a double colon (::) for linkages that fuse open reading frames. The following worm strain was generated and used in this study: unc-119(ed3) III; curIs23 [Pcdh-3 > Lifeact::Dronpa-M159T + unc-119(+)].

The DronpaM159T sequence was obtained by adapting the mammalian sequence for C. elegans codon usage and putting in syntrons using the website http://worm-srv3.mpi-cbg.de/codons/cgi-bin/optimize.py. Lifeact and linker were synthesized in frame with DronpaM159T[32] by Eurofins Genomics. Lifeact::Dronpa-M159T was then recombined into destination vector pCFJ150 with the promoter cdh-3 and the 3′UTR of unc-54 via Gateway cloning giving the construct pCFJ150-Pcdh-3 > Lifeact::Dronpa-M159T. 30 ng/μl of the construct was coinjected with DNA 1 Kb ladder as a carrier at up to 100 ng/μl in unc-119(ed3) worms. Stably expressed extrachromosomal lines were selected and integrated by UV irradiation using the Stratagene UV Stratalinker 2400.

**Reporting summary**. Further information on experimental design is available in the Nature Research Reporting Summary linked to this article.

**Code availability**. The custom software used in this study is available from the corresponding authors upon reasonable request.

## Data availability

The data that support the findings of this study is available from the corresponding authors upon reasonable request.

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

## Acknowledgements

We thank the European ERC starting grant "MoNaLISA" funding https://doi.org/10.13039/501100000781 for supporting the project, Dr. Sami Koho and Giorgio Tortarolo for useful discussions. J.D. thanks the Carlsberg Foundation for financial support. G.C. and I.T. thank the Swedish Research Council for funding. J.P. acknowledges financial support from the Fondation ARC (Grant Number PJA 20151203487). G.V. acknowledges financial support from the Compagnia di San Paolo (codice ROL 20641). This work also received support under the program «Investissements d'Avenir» launched by the French Government and implemented by ANR with the references ANR-10-LABX-0038 and ANR-10-IDEX-0001-02 PSL, including financing of R.C.'s short-term stay in Sweden. R.C. was funded by a Ph.D. fellowship from ITMO Cancer.

## Author contributions

I.T. and G.V. conceived the project idea and supervised research. G.V. and M.C. conceived the idea and implemented the algorithms and the architecture of the smart pixel technology. I.T. and J.D. developed the imaging platforms and acquired data. G.C. prepared sample and biological guidance for organelles live-cell imaging. J.P. and R.C. provided biological guidance and the C. elegans strain and R.C. acquired data. All authors analyzed data. I.T., J.D., and G.V. wrote the manuscript with input from all the authors.

## Additional information

**Competing interests:** The authors declare no competing interests.

