## [Peer Review File · Nature Communications]

REVIEWERS' COMMENTS:

Reviewer #1 (Remarks to the Author):

The authors addressed all concerns carefully and appropriately in their response to reviewers. If they add the discussion about possible photodamage effects and spatial resolution to the manuscript not only in the response to the reviewer I am strongly supporting publication of the revised manuscript.

Response Letter to the reviewers comments

We thank the editor and reviewer for the valuable comments.

In the next section we provide a point-by-point response. The reviewer comments are in blue, our responses are in black and any additions to the main text and supplementary information is cited and explained below.

Reviewer #1 (Remarks to the Author):

The authors addressed all concerns carefully and appropriately in their response to reviewers. If they add the discussion about possible photodamage effects and spatial resolution to the manuscript not only in the response to the reviewer I am strongly supporting publication of the revised manuscript.

We thank the review for considering our revised version of the paper suitable for publication.

We added the discussion on phototoxicity as a supplementary note 10 and we cited this discussion in the main text.

Additionally, we added few sentences referring to the spatial resolution and FRC and we clarified when the mentioned value refers to FWHM or FRC analysis. We also added a supplementary figure showing the FRC results, which is mentioned in the main text as well.

Finally, we are willing to engage in further discussion if some points remain unclear.